# Evaluation of Tensile Bond Strength between Self-Adhesive Resin Cement and Surface-Pretreated Zirconia

**DOI:** 10.3390/ma15093089

**Published:** 2022-04-24

**Authors:** Mijoo Kim, Reuben H. Kim, Samuel C. Lee, Thomas K. Lee, Marc Hayashi, Bo Yu, Deuk-Won Jo

**Affiliations:** 1Restorative Materials and Applied Dental Research Laboratory, UCLA School of Dentistry, Los Angeles, CA 90095, USA; vagusmj@gmail.com (M.K.); rkim@dentistry.ucla.edu (R.H.K.); tlee@dentistry.ucla.edu (T.K.L.); mhayashi@dentistry.ucla.edu (M.H.); boyu@dentistry.ucla.edu (B.Y.); 2Section of Restorative Dentistry, UCLA School of Dentistry, Los Angeles, CA 90095, USA; 3California Smile Dental Studio, Gardena, CA 90247, USA; samclee28@gmail.com; 4Department of Prosthodontics, Section of Dentistry, Seoul National University Bundang Hospital, Seongnam 13620, Korea

**Keywords:** acid etching, self-adhesive resin cement, surface treatment, tensile bond strength, zirconia, zirconia nanoparticle coating

## Abstract

The tensile bond strength between zirconia subjected to different surface-pretreatment methods and methacryloyloxydecyl-dihydrogen-phosphate (MDP)-containing self-adhesive resin cement was evaluated herein. Eighty-eight cylindrical zirconia specimens were randomly divided into the following four groups based on the pretreatment method: (1) no treatment, (2) air abrasion, (3) HNO_3_/HF etching, and (4) zirconia-nanoparticle coating. The tensile bond strength of the zirconia–resin-cement complexes was investigated. One-way ANOVA and post hoc tests were performed at a 95% significance level, and the Weibull modulus was calculated. Fracture patterns were visualized by SEM. The surface roughness of the specimens without resin bonding was evaluated by AFM. The tensile bond strength of the specimens decreased as follows: Groups 3 > 4 > 2 > 1 (28.2 ± 6.6, 26.1 ± 5.7, 16.6 ± 3.3, and 13.9 ± 3.0 MPa, respectively). Groups 3 and 4 had significantly higher tensile bond strengths (*p* < 0.05) and lower fracture probabilities than those of Groups 1 and 2. They also showed both mixed failure and resin-cement cohesive failure, whereas Groups 1 and 2 showed mixed failure exclusively. The zirconia–resin tensile bond was stronger after HNO_3_/HF etching or ZrO_2_-nanoparticle coating than after air abrasion or no treatment. The estimated surface roughness decreased as follows: Groups 3 > 4 > 2 > 1. The combination of zirconia pretreated with HNO_3_/HF etching or ZrO_2_-nanoparticle coating and an MDP-containing self-adhesive resin cement can increase the clinical longevity of zirconia restorations by preventing their decementation.

## 1. Introduction

Zirconia is extensively used for dental restorations because of its superior mechanical properties and biocompatibility. Until recently, it suffered from two major shortcomings: it lacked translucency and was associated with a high incidence of secondary caries and marginal discoloration due to mismatches between the zirconia framework and chemically-unbonded luting [1,2]. However, the recent development of more esthetically appealing, highly translucent monolithic zirconia has made zirconia restorations a popular alternative to porcelain-fused-to-metal ones [3,4]. Furthermore, monolithic zirconia exhibits a high fracture resistance and a decent marginal fit, which can assist in cases with reduced occlusal thickness and help preserve more tooth structure [5,6]. As manufacturing technology has advanced, clinical studies have shown high success and survival rates for tooth-supported monolithic-zirconia restorations [4,7], as well as acceptable internal and marginal fits [6].

Nevertheless, factors such as preparation and design, scanning, computer-aided design and manufacturing (CAD/CAM) procedures, and the zirconia material itself can still potentially compromise the fit of zirconia restorations [5,8,9]. Moreover, tensile-stress concentrations can theoretically occur at the inner surfaces of the restorations under loading, because dentin and zirconia have different elastic moduli [10,11,12,13]. This could lead to mechanical failure and subsequent bacterial invasion. Therefore, chemical bonding using resin cement with an adequate elastic modulus and tensile strength has been employed to reduce the risk of such complications [14].

Researchers have attempted to achieve optimal zirconia–resin bond strengths by addressing the chemical inertness of zirconia [15,16]. Many chemical and mechanical pretreatment methods for zirconia surfaces have been actively studied, including the use of 10-methacryloyloxydecyl dihydrogen phosphate (10-MDP) [17,18,19,20], air abrasion [21], hydrofluoric acid (HF) etching [22,23], tribochemical silica coating [24,25], and ceramic coating [26]. However, an optimal resin bonding method for zirconia has not yet been established.

The 10-MDP-based method is currently believed to be the most appropriate strategy for increasing the zirconia–resin bonding strength [17,18,19,20]. Therefore, the use of MDP-containing self-adhesive resin cement is considered to be a promising approach toward improving the chemical bonding, retention, and stability of zirconia restorations. The use of surface abrasion for zirconia surfaces has also been extensively studied, but its influence on zirconia–resin bonding has not been sufficiently clarified: several studies have reported conflicting results on the increase and decrease in bond strength after sandblasting [27,28,29].

Other zirconia pretreatment methods, such as nitric-acid/hydrofluoric-acid (HNO_3_/HF) etching [30,31,32] and zirconia-particle coating [33,34], have been developed to improve the zirconia–resin bond strength by increasing the zirconia surface roughness. Acid etching enables the homogenous roughening of material regardless of its size and shape [35]. Nanoparticle technology based on a delivery carrier has been developed to improve the mechanical, physical, and chemical properties of the coatings and effectively transport materials to the target area [36,37]. A nanoparticle coating method for zirconia surfaces was developed by employing carbon films through hydrogen bonding; the carbon particles produced reticulated porous zirconia after sintering [38]. However, information on the comparison of different treatment methods is limited. Therefore, the present study seeks to determine the effect of pretreating a zirconia surface by HNO_3_/HF etching or ZrO_2_-nanoparticle coating on the zirconia–resin-cement tensile bond strength. The null hypothesis is that the bond strength is not affected by the different pretreatment methods.

## 2. Materials and Methods

### 2.1. Specimen Preparation

The materials used in this study are listed in Table 1. Yttria-partially stabilized tetragonal zirconia (Y-TZP) blocks (ADD-Z shade block; PNUADD, Busan, Republic of Korea; Yttria concentration of 4%) were cut into 88 cylindrical shapes to fabricate specimens with final dimensions of 3 mm × 10 mm (diameter × length) after sintering. The cut surfaces were polished with 600-grit silicon carbide under running water. The cylinders were randomly assigned to Groups 1–4, and treated as follows:Group 1: not treated;Group 2: air-abraded with 50-μm-sized alumina particles under a pressure of 0.1 MPa from a distance of 10 mm for 20 s and at incidence angles of 60–90°;Group 3: etched using Zircos-E HNO_3_/HF etching solution;Group 4: treated with ZirADD zirconia-nanoparticle coating.

Groups 1–3 were sintered at 1530 °C for 12 h before the different surface treatments; however, Group 4 was first subjected to surface coating according to the manufacturer’s guidelines and was then sintered.

The Zircos-E etching solution (M&C Dental, Seoul, Korea) for the Group 3 specimens was used in a working box to ensure safety. A mixture of nitric acid and hydrofluoric acid in equal ratios was prepared and buffered for surface treatment according to the manufacturer’s datasheet. The zirconia specimens were treated for 2 h and immersed in an ultrasonic cleaner for 30 min. Subsequently, the zirconia cylinders were cleaned with cold running water and steam and then subjected to the following annealing procedure: they were heated to 1150 °C, maintained at that temperature for 30 min, and cooled until the temperature was below 200 °C.

The ZirADD slurry (PNUADD, Busan, Korea) used for the Group 4 samples was spread once evenly onto the bonding surface of the zirconia cylinders prior to sintering; the thickness of the coating layer was approximately 3–4 μm, as indicated in the manufacturer’s guidelines. The cylinders were then placed in a furnace and sintered at 1530 °C for 12 h.

### 2.2. Bonding Procedure

After performing the pretreatment procedures on each zirconia specimen, a surface cleaning agent (ZirClean; Bisco Inc., Schaumburg, IL, USA) was applied to the zirconia bonding surface for 20 s, washed, and dried with oil-free air. Each zirconia specimen was placed in a putty mold with a cylindrical hole 3 mm in diameter and 20 mm in length. Two-millimeter-long MDP-containing self-etching/self-adhesive resin cement (Theracem; Bisco Inc., Schaumburg, IL, USA) was incrementally squeezed onto the zirconia specimen and light-cured for 20 s using a variable-intensity polymerizer (Bisco Inc., Schaumburg, IL, USA; 600 mW/cm^2^). This procedure was repeated five times, eventually yielding 10-mm-long resin cement. After the bonding procedure, all specimens were stored in distilled water at 37 °C for 24 h prior to the tensile bond tests.

### 2.3. Tensile-Bond Test and Weibull Distribution

The specimens were examined using a light stereomicroscope (SE303R-P; AmScope, Irvine, CA, USA) at 10× magnification to identify defective specimens with air bubbles or interfacial gaps. Each sample was fixed to the test jaws using cyanoacrylate cement (Zapit; DVA, Corona, CA, USA); the jaws could be moved in opposite directions in the microtensile tester (Bisco Inc., Schaumburg, IL, USA). The two meeting jaws possess a 2-mm-long notched space for the application of tensile strength at the interface between zirconia and resin cement. The tests were performed at a crosshead speed of 1.0 mm/min. The tensile bond strength (MPa) was obtained by dividing the recorded peak load at failure (N) by the adhesive surface area (mm^2^).

Weibull statistics were performed following the ISO 6872, Annex B protocol. The strength distributions of quasi-brittle materials, such as ceramics, are more appropriately described by Weibull statistics than the mean strength values determined using a Gaussian strength distribution. The Weibull modulus (*m*) was used to assess the variability in strength; a smaller *m* indicates a lower reliability of the strength. The fundamental equation for the Weibull distribution is
(1)$$P_{f}=1-exp\left[-{\left(\frac{\sigma }{\sigma_{0}}\right)}^{m}\right],$$
where *P_f_* is the probability of failure, *σ* is the stress applied during tensile testing, and *σ*_0_ is the Weibull characteristic strength corresponding to a failure probability of 63.21%. The Weibull distribution can be written in the form of a linear equation
(2)$$lnln\left[\frac{1}{1-P_{f}}\right]=mln\left(\frac{\sigma }{\sigma_{0}}\right);$$
the slope of the straight line fitting the data to this equation was determined to find *m*.

### 2.4. Failure-Mode Evaluation

The failure modes were analyzed using an optical microscope at a magnification of 30× after the tensile bond tests. Three types of failure modes were analyzed: (i) adhesive failure between the zirconia and resin cement, (ii) cohesive failure, and (iii) a mixed mode featuring both.

### 2.5. Scanning Electron Microscopy (SEM) Analysis

After the tensile bond tests, the zirconia specimens were mounted on aluminum stubs, sputter-coated, and examined by SEM (JEOL 5600 LVj; JEOL Ltd., Tokyo, Japan). A 10-nm-thick gold layer was coated onto the specimens using a sputter coater (BAL-TEC SCD 005 Sputter Coater; Balzers, Liechtenstein) to impart electrical conductivity. An accelerating voltage of 5 kV was used. SEM images were obtained three times for each group in specific areas of interest at various magnifications.

### 2.6. Atomic Force Microscopy (AFM) Analysis

Cubic zirconia specimens (5 × 5 × 10 mm^3^) subjected to the different surface pretreatments were prepared and analyzed with an atomic force microscope (MFP-3D; Asylum Research, Santa Barbara, CA, USA). A single operator analyzed the average surface roughness (Ra) of each group, which was expressed as a numeric value (nm) via Nanoscope V530R35R software. Images were captured in the air. Fields of view at a scan size of 10 µm × 10 µm were considered and recorded at a slow scan rate (0.1 Hz).

### 2.7. Statistical Analysis

The tensile bond strength of the zirconia-based specimens in Groups 1–4 was analyzed using SPSS software (version 26.0; SPSS Inc., Chicago, IL, USA). One-way analysis of variance (ANOVA) and post hoc tests were performed at a 95% significance level.

## 3. Results

The mean bond strengths (±SD) of the samples in Groups 1–4 were 13.9 ± 3.0, 16.6 ± 3.3, 28.2 ± 6.6, and 26.1 ± 5.7 MPa, respectively (Figure 1). The Group 2 specimens (air abrasion) did not show a significantly enhanced bond strength compared to that of the control (*p* > 0.05). However, samples in Groups 3 and 4 (HNO_3_/HF etching and zirconia-nanoparticle coating, respectively) showed considerably higher bond strengths than did the samples in the control group (*p* < 0.05); moreover, Groups 3 and 4 were not significantly different (*p* > 0.05). In the Weibull plots of the fracture data (Figure 2), the values of *m* for Groups 1–4 were 5.56, 6.07, 5.02, and 5.44, respectively. Group 3 had the lowest probability of bonding fracture, followed by Groups 4, 2, and 1.

Eighty-two out of the eighty-eight specimens showed the mixed failure mode (adhesive and cohesive), as determined observationally using a light microscope (Figure 3). All of the specimens in Groups 1 and 2 exhibited the mixed failure mode, whereas 23% and 5% of the samples in Groups 3 and 4, respectively, demonstrated notches corresponding to cohesive failure. Given the substantial fracture toughness of zirconia, this must represent fracture within the resin cement.

SEM analysis (Figure 4) revealed variations in the mixed failure modes of the various groups. The Group 4 zirconia specimen, which was treated with the ZirADD nanoparticle coating, was shown to be coated with ZrO_2_ particles (Figure 4d,h). After the tensile bond tests, the Group 3 sample showed a uniformly etched zirconia surface (Figure 4g). The AFM results were in line with the SEM images. The HNO_3_/HF-etched sample (Figure 5c) showed the highest Ra value (nm) with a flatter height distribution than that of the normal pattern (kurtosis Rku < 3) among the treated groups, as shown in Figure 4c,g. The mean value of Ra was the highest for the Group 4 sample (Figure 5d); the other untreated or treated groups had considerably more even topographies, as shown in Figure 5.

## 4. Discussion

The effects of the HNO_3_/HF-etching and ZrO_2_-nanoparticle-coating methods on the zirconia surface were investigated in this study by examining the tensile bond strength between zirconia and resin cement. Significant differences in bonding performance compared to that of the control group were observed only in the acid-etched and zirconia-nanoparticle-coated groups; therefore, the null hypothesis was partially rejected.

The tensile bond strength test was performed to analyze the fracture surfaces after debonding. Despite the lack of consensus on the most appropriate bond-strength-testing protocols for investigating zirconia–resin complexes, the tensile bond test is a verified and widely used method [39]. Furthermore, the execution of this test, including the fabrication of uniform specimens, is straightforward. The microtensile tester used in this study has the advantage of reducing the effects of gravity because the tensile force is exerted horizontally, unlike that in a typical universal testing machine with vertical jigs.

The MDP-containing self-adhesive resin cement was selected as a bonding substrate because it is preferred in clinical settings. Clinicians are generally more likely to avoid labor-intensive and multistep procedures in daily practice, even if such procedures are more reliable. To achieve an optimally strong dentin–ceramic bond, multiple procedures such as isolating the tooth from moisture and saliva, acid etching/washing/drying, applying a primer/adhesive, using dual-curing resin cements and the curing mode of the resin luting material, and using an oxygen-inhibiting gel depending on the luting system are necessary; all of these procedures are conducted while restricting patient movement. Complicated steps imply an increased possibility of multiple errors in each stage and could eventually lead to clinical failure.

MDP is known to accelerate the bonding strength with zirconia via ionic and hydrogen bonds upon being added to the resin cement or as a primer. A two-year follow-up study has revealed that MDP yielded successful clinical outcomes without biological or technical complications [3]. Further, the application of an MDP-containing agent based on organophosphate/carboxylic acid monomers (Z-prime Plus; Bisco Inc., Schaumburg, IL, USA) has been found to increase the bond strength of different resin-based luting agents, including the Z100 restorative material [18]. Moreover, a similar zirconia–resin shear bond strength was reported for the combined use of an MDP primer with an MDP-free self-adhesive resin cement and the use of an MDP-containing resin cement alone [20]. In the present study, the Group 1 samples (control group) did not exhibit an adhesive failure mode and had a tensile bond strength of 13.9 ± 3.0 MPa, similar to that obtained in a macrotensile bond test on tribochemical-silica-coated zirconia [31] and higher than that of other MDP-containing adhesive materials [33].

Although air abrasion is an extensively used zirconia-pretreatment method, there is no consensus on experimental conditions such as the Al_2_O_3_ particle size, air-pressure setting, duration, and incident angle. Özcan et al. performed air abrasion using alumina particles (Ø 30–50 μm) at a pressure between 0.5 and 2.5 bar (0.05 and 0.25 MPa in SI units) for at least 20 s [34]. The blast jet was positioned 10 mm from the target and kept in motion to prevent the creation of defects. In another study, the specimens were sandblasted using 150-μm-grain-sized aluminum oxide particles for 20 s at a pressure of 3.8 bar (0.38 MPa in SI units) with a tip distance of 10 mm from the ZrO_2_ blocks and perpendicular to their surface, which significantly increased the microshear bond strength compared to that of the non-treated group [26]. The present study followed the air-abrasion protocol established by Pontevedra et al., who demonstrated successful clinical outcomes after two years [3]. The effects of air abrasion combined with MDP have also been investigated. Yang et al. showed that air abrasion resulted in a durable tensile bond to zirconia even at a reduced abrasion pressure when combined with MDP-containing primers [40]. However, Shahin et al. concluded that the increase in retention by adhesive resins was considerably greater than the effect of air abrasion [41]. The present study found that the air-abrasion method combined with an MDP-containing self-adhesive cement increased the tensile bond strength compared to that of the control (*p* > 0.05).

HF is known for its ability to dissolve silicon oxide, the main ingredient in glass. Essentially, HF can dissolve the glassy phase, leaving behind the crystalline phase and creating surface roughness. Previously, zirconia, which lacks a glassy phase, was believed to not react with HF. However, control of the HF concentration, immersion time, and temperature can increase the surface roughness of zirconia specimens and induce tetragonal-to-monoclinic transformation [22]. Moreover, HNO_3_/HF etching significantly roughens the zirconia surface [30]. In the present study, this surface-treatment method increased the tensile bond strength and roughened the zirconia surface, as shown in Figure 1, Figure 4 and Figure 5. Based on these results alone, HNO_3_/HF etching might seem to be the most effective treatment method for zirconia. However, these acids are extremely corrosive and hazardous materials; their fumes can threaten the health of people in dental laboratories.

Considering the risks and benefits of the various treatment methods, the ZrO_2_-nanoparticle coating strategy may be the most effective option. Because of the carbon powder, the coating is initially black, which enables a recognizable uniform layer to be applied; the esthetically undesirable black color disappears after sintering. The handling process is simple and safe because of the lack of harmful emitted vapors. Liu et al. showed that zirconia-particle coating led to a shear bond strength superior to that of tribochemical silica coating, glazing porcelain coating after sintering of zirconia, and silica slurry coating prior to the sintering of zirconia after thermocycling [42]. The increase in Ra in the present study from 60.08 nm (control) to 164.765 nm shows that use of the slurry produces excellent results. However, caution is needed when using this pretreatment method in practice, because clinical try-in procedures after sintering may damage the nanoparticle-coated inner surface. Minor modifications of the intaglio surface prior to cementation can affect the bond strength between the tooth and the zirconia restoration.

To the best of our knowledge, this study is unique in comparing different commercially available zirconia surface treatments and analyzing the bonding force between the surface-treated zirconia and the resin cement. However, aging conditions were not considered in this study. The incorporation of thermocycling and water storage to mimic a clinical setting could weaken the bond strength owing to the hydrolysis of the polysiloxane network between the ceramic substrate and the polymerized intermediate resin [43,44]. Therefore, the effects of aging conditions on zirconia–resin bonding using these pretreatment methods should be evaluated in further studies.

## 5. Conclusions

Surface treatments based on HNO_3_/HF etching and ZrO_2_-nanoparticle coating strengthened the tensile bond of zirconia to MDP-containing self-adhesive resin cement compared to that observed in untreated or air-abraded specimens. The roughest surface was observed in the HNO_3_/HF group, followed by those in the zirconia-nanoparticle-coated, air-abraded, and untreated groups. According to the results obtained in this study, the combination of zirconia pretreated with HNO_3_/HF etching or ZrO_2_-nanoparticle coating and an MDP-containing self-adhesive resin cement can increase the clinical longevity of zirconia restorations by improving the surface roughness and tensile bond strength.

## Figures and Tables

**Figure 1 materials-15-03089-f001:**
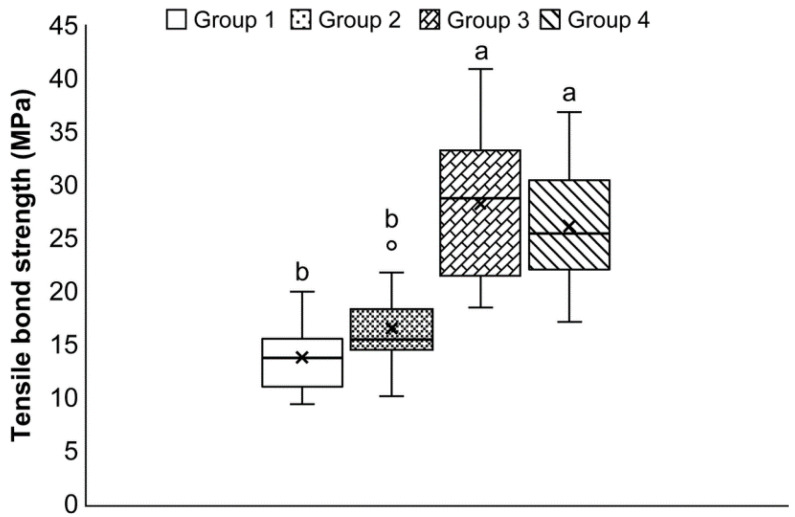
Box plot of shear bond strength (MPa) of samples belonging to Groups 1–4. The different letters represent significant differences between the groups (*p* < 0.05).

**Figure 2 materials-15-03089-f002:**
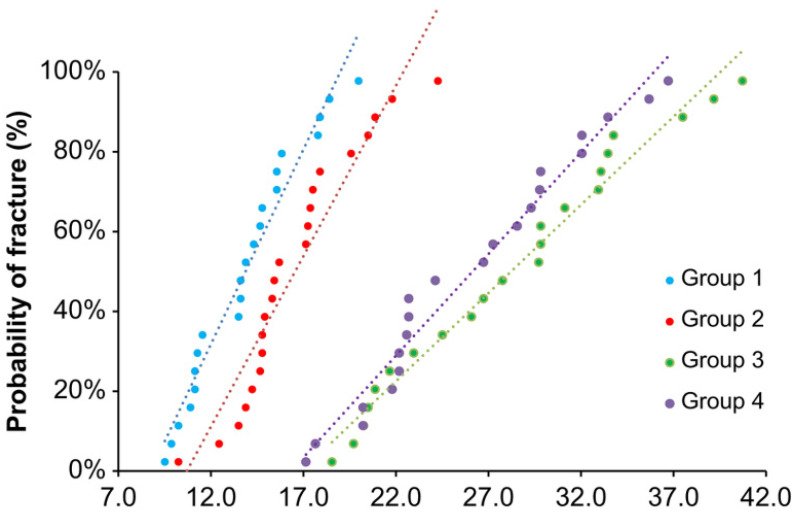
Weibull representation of the fracture data of the Group 1–4 samples.

**Figure 3 materials-15-03089-f003:**
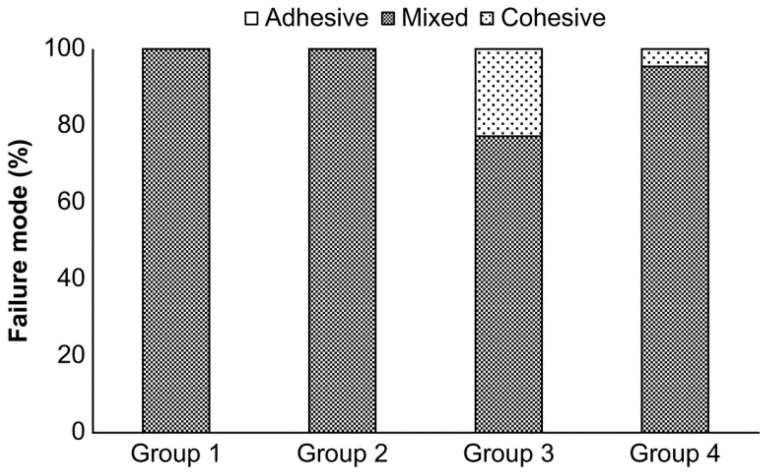
Failure mode analysis of the Group 1–4 samples.

**Figure 4 materials-15-03089-f004:**
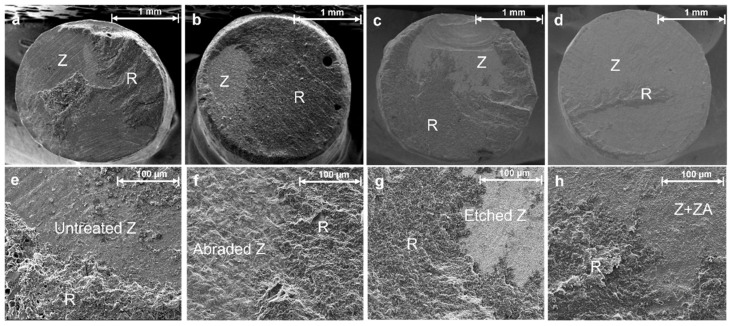
Scanning electron microscopy (SEM) images showing the mixed failure modes in Groups (**a**,**e**) 1, (**b**,**f**) 2, (**c**,**g**) 3, and (**d**,**h**) 4. Each pair of images was acquired at 50× and 1000× magnifications, respectively. Z, R, and ZA in the images represent zirconia, resin cement, and the ZirADD coating, respectively.

**Figure 5 materials-15-03089-f005:**
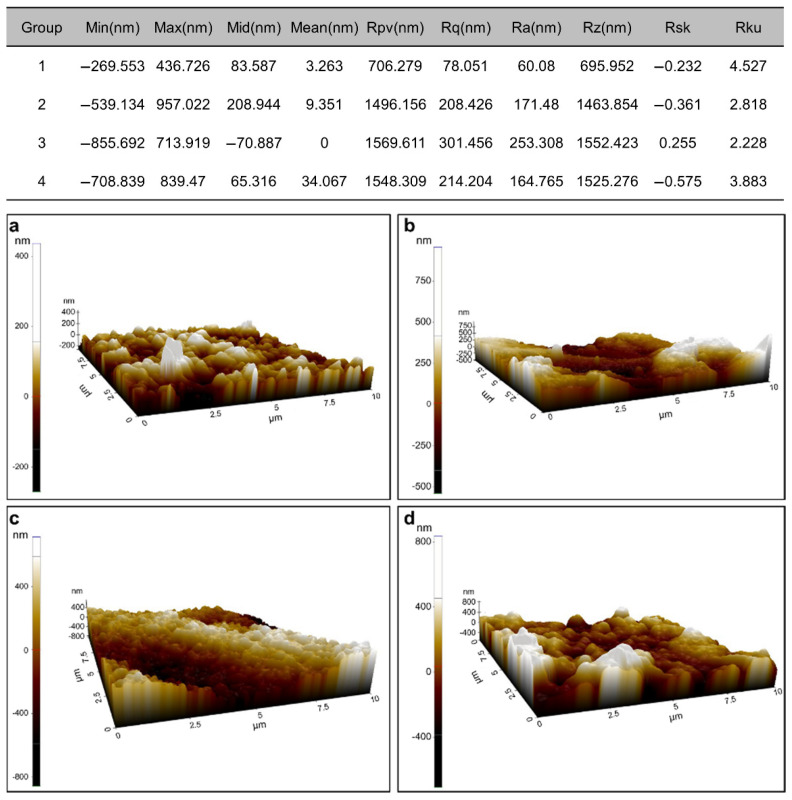
Representative atomic force microscopy (AFM) images of specimens belonging to various groups: (**a**) 1 (control), (**b**) 2 (air abrasion), (**c**) 3 (Zircos-E etching), and (**d**) 4 (ZirADD coating).

**Table 1 materials-15-03089-t001:** Material specification.

Product	Manufacturer	Composition
ZirClean cleaning agent	Bisco Inc., Schaumburg, IL, USA	Zirconium oxide, water, polyethylene glycol, potassium hydroxide, pigments, additives
Theracem resin cement	Bisco Inc., Schaumburg, IL, USA	Calcium base filler, silanated non-reactive fillers, methacrylate monomers containing phosphoric acid groups, methacrylate monomers, ytterbium fluoride, initiators (chemical and light)
Zircos-E etching solution	M&C Dental, Seoul, Korea	Hydrofluoric acid and nitric acid
ZirADD zirconia-nanoparticle coating	PNUADD, Busan, Korea	Distilled water, nano-sized zirconia powder, carbon powder, dispersive agent, binder

## Data Availability

Not applicable.

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
