# Peer review of "Evaluation of Tensile Bond Strength between Self-Adhesive Resin Cement and Surface-Pretreated Zirconia"

_materials, 2022, doi:10.3390/ma15093089_

Round 1
Reviewer 1 Report
I would like to thank the authors for their great work. The article is well structured, correctly written and presents a correct study design.
I just have a few small observations for the authors, with the aim of making the article more easily understandable.
- Line 45. Sentence " ...as well as acceptable internal and marginal fits." needs a reference. If there is no such information available in the scientific literature, the sentence should be removed.
- Line 111. Which kind of water was used? Physiological Solution, Distilled water, Artificial saliva?
- Lines 222-225. I believe that such an important statement, however true and irrefutable, must be supported by some bibliographic references. A suggestion on this could be DOI: 10.1016/j.ijadhadh.2020.102673
- Lines 232-235. In my opinion it is important to mention the importance of using a dual-curing resin cements and the curing mode of the resin luting material.
Reviewer 2 Report
The manuscript entitled “Evaluation of tensile bond strength between self-adhesive resin cement and surface-pretreated zirconia” evaluated the effects of surface treatment on the tensile bond strength between zirconia and self-adhesive resin cement. The authors concluded that HNO3/HF etching or ZrO2 nanoparticle coating can increase the tensile bond strength. These results are interesting and useful for zirconia restorations. Following points, especially descriptions for Materials and Methods, should be revised before acceptance.
P2/ L78 The yttria concentration of Y-TZP should be described.
P2/ L85 “1 bar” is better to convert to SI unit (MPa)
P2/ L90 Surface treatment processes for Group 1, 2 and 4 are not clear. Did all specimens were subjected to annealing procedure? It is better to describe how to prepare surface treatment for each Group.
P3/ L97 Please describe approximately thickness of nanoparticle coating layer.
P3/ L100 Table 1 Information of materials is insufficient. Concentrations of potassium hydroxide, hydrofluoric acid and nitric acid should be described. Please try to check and describe the compositions of ZirADD zirconia-nanoparticle coating.
P3/ L119 Please describe more detailed set condition of microtensile tester, such as the distance between two jaws and where the samples were fixed with Zapit.
P3/ L141 P5/ L187 Is it true to show cohesive failure of zirconia? If it occurred, which part of zirconia was fractured, within nanoparticle layer or between nanoparticle and zirconia cylinder? If the cohesive failure of zirconia did not occur, please describe as cohesive failure of the resin cement.
P8 L290 The modifications of the intaglio surface influence the fitting of the restoration. Please describe this disadvantage.
Reviewer 3 Report
The manuscript is focused on the use of tensile bond strength between self-adhesive resin cement and surface-pretreated zirconia.
The topic is appropriate for the journal.
The title is adequate and correlate with the content of the article.
The abstract reports a consistent summary of the article findings.
The work has a clear structure.
All sections are properly written and required for a complete understanding.
Nevertheless, there are minor issues that require to be addressed before proceeding with the publication, to enhance the quality and presentation to a broad audience.
A language, typos and punctuation check may be useful.
Furthermore, the manuscript could strongly benefit of additional references over section “1.2 Nanoparticles transport mechanism” as mentioning a more complete overview of the attempted approaches would emphasize the scientific soundness of the presented findings (Wang et al., Beilstein J. Nanotechnol., 2018, 9, 137–145; Vallet-Regì et. al., Molecules, 2018, 23(1): 47).
Fig. 5 interestingly reports on the AFM 3D: the manuscript would increase its scientific importance whether it could be possible to add on the related AFM height profile graphs.
The conclusion section would definitely benefit futher explanation, e.g. addition of a few sentences recapitulating the whole findings, the scientific progress and soundness of the original research work.
